# Sustainable Composites from Nature to Construction: Hemp and Linseed Reinforced Biocomposites Based on Bio-Based Epoxy Resins

**DOI:** 10.3390/ma16031283

**Published:** 2023-02-02

**Authors:** Julio Vidal, David Ponce, Alice Mija, Monika Rymarczyk, Pere Castell

**Affiliations:** 1Aitiip Centro Tecnológico, Research and Development Department, 50720 Zaragoza, Spain; 2Institute of Chemistry of Nice, University Côte d’Azur, UMR CNRS 7272, CEDEX 02, 06108 Nice, France; 3Centexbel-VKC, Technologiepark 70, Zwijnaarde, 9052 Gent, Belgium

**Keywords:** linseed, hemp, biocomposites, bioepoxy, non-woven reinforcements

## Abstract

The present manuscript describes the use of natural fibers as natural and sustainable reinforcement agents for advanced bio-based composite materials for strategic sectors, for example, the construction sector. The characterization carried out shows the potential of both natural hemp and linseed fibers, as well as their composites, which can be used as insulation materials because their thermal conductivity properties can be compared with those observed in typical construction materials such as pine wood. Nevertheless, linseed composites show better mechanical performance and hemp has higher fire resistance. It has been demonstrated that these natural fibers share similar properties; on the other hand, each of them should be used for a specific purpose. The work also evaluates the use of bio matrixes in composites, demonstrating their feasibility and how they impact the final material’s properties. The proposed bio-resin enhances fire resistance and decreases the water absorption capacity of the natural fibers, enabling the use of composites as a final product in the construction sector. Therefore, it has been demonstrated that it is possible to manufacture a biocomposite with non-woven natural fibers. In fact, for properties such as thermal conductivity, it is capable of competing with current materials. Proving that biomaterials are a suitable solution for developing sustainable products, fulfilling the requirements of the end-user applications, as it has been demonstrated in this research with the non-woven fibers for the non-structural components.

## 1. Introduction

The European Commission published the Green Deal, a document that set several environmental issues and goals and is being used by European countries to move towards a more sustainable and circular economy [1], in December 2019. Among all the different facts and references published in it, the Commission’s document shows that the construction sector is responsible for 20% of European emissions of greenhouse gases. Furthermore, with transport and energy sectors, this sector all together produces more than 50% of the total greenhouse emissions in Europe. Therefore, to reach the CO_2_ neutrality set for 2050 and the reduction of 55% of CO_2_ emissions by 2030, it is essential to tackle the environmental problem in these sectors by adapting their activities and increasing their environmental sustainability [2,3,4]. Moreover, the pandemic effects and consequences and the current problem of the global supply chain have uncovered the need to find new sources of materials.

Among the different materials options that must be studied and improved in these sectors, composites are a perfectly suitable alternative to help achieve the environmental goal in all the above sectors. A composite material is formed by two or more materials, achieving a performance that would not be achieved by any of these components by themselves. This material is composed of reinforcement, a material that enhances mechanical properties, and a matrix, which is the component that consolidates all the reinforcement together through good reinforcement impregnation. The use of natural components in construction is not a novelty. The Egyptian civilization used composite materials reinforced with natural fillers in their constructions [5]. For example, they used straw fibers.

Nowadays, the construction sector also uses composites in some situations. Glass fibers are being used as reinforcement to obtain more resistant and insulating materials [6,7,8,9], allowing the reduction of energy used to maintain a constant and comfortable temperature inside the buildings. Considering the European market, there are two potential sustainable solutions to substitute glass fiber in composites: natural hemp and/or linseed fibers. They are the main alternatives due to their local availability and properties. For this reason, they have obtained the interest of many researchers and industries [10,11,12,13,14].

Moreover, the implementation of natural fibers as reinforcement in composite materials is a valuable and strategic opportunity for the European agricultural sector. The use of hemp and linseed fibers would allow their revalorization while reducing waste production, reaching the goals established at the international level and moving towards a zero-waste goal. Nowadays, tons of natural fibers are being produced and wasted each year [15]; therefore, their revalorization could have a huge potential and could bring a real economic impact to rural agricultural regions in Europe, achieving one of the objectives of the Green Deal strategy. The use of these kinds of resources would imply the generation of new economic opportunities, such as their use in the construction sector.

Several studies have already proved the great potential of natural fibers [16] to be used as reinforcements [17,18], given that the mechanical properties of the composites which incorporate them can reach similar outcomes to those composites of glass fiber [19,20,21,22]. These results open many market niches for natural fibers, but because their cost and reliability are still under-evaluated, they are found mainly in non-structural parts and less demanded applications or products [23]. 

Therefore, there are some challenges and technical limitations that must be overcome for the use of vegetal fibers at the same level as the current reinforcement materials. As reported in previous research, the adherence between the matrix and the reinforcement is a critical requirement to maximize the materials’ potential and their performance [24]. A review conducted by Baley et al. [25] considered how to boost the mechanical performance of the fiber itself. The use of binders and/or the modification of resins or fibers can be required to reinforce the interactions between them, so the final properties of the material are achieved at the commercial level. Another alternative to the use of binders would be the treatment of the surface of the fiber, as proposed by Obame et al. [26], which explained how, through the use of sodium hydroxide, they improved the fiber-matrix adhesion. 

Thus, to achieve the circular economy premise for the whole composite system, there is not only the need to focus on the fibers but also on the resins. The aim of this study was to optimize the resin formulation to improve its adherence to the fibers in order to enhance product sustainability, carbon emissions and environmental footprint while taking biocomposites a step forward for their application into the market through the use of standardized characterizations. In this sense, plenty of work shows the big potential of natural fibers and resins and how their nature, structure and surface treatments impact the final properties of the composites [27]. Some of these researchers have not stopped at the mechanical performance or the chemical properties, but they have taken into consideration the environmental impact and the future of these newly developed composites [28,29].

One of the most common types of thermosets in the construction sector is epoxy resins [30,31,32]. The final thermoset chemical structure and properties are governed by the chemical crosslinking, as is the monomer vs. hardener/initiator reactivity and compatibility [33,34]. Therefore, in order to obtain composites that allow carbon footprint reduction, new epoxy resins with high bio-based organic carbon content must be developed [9,10].

The chemical industry is moving towards the great variety of molecules that can be found in nature as a source of raw materials in the production of thermoplastics, thermosets and other products with industrial use [35]. One of the most abundant and widely used resources for the preparation of bio-epoxy resins are unsaturated vegetable oils such as castor oil, linseed oil, soybean oil and cardanol oil. Their epoxidation by simple and green oxidation of fatty acids unsaturation allows the synthesis of aliphatic bio-based epoxy monomers [36]. In this study, epoxidized linseed oil (ELO) has been selected due to its high functionality (~5.5 epoxy groups/triglyceride) and the large accessibility of linseed crops in Europe. In the last years, many efforts have been focused on the sustainable synthesis of ELO-based thermosets [33,34,37]. Moreover, biomolecules that draw plenty of attraction are sugars [38] and biopolymers such as cellulose [39] and lignin [40]. This interest is due to their properties and chemical structures because these natural compounds can be used as raw materials for the fabrication of matrices. Furthermore, in the case of lignin, lignin-based carbon fibers can be generated [41] and combined with lignin-based resins [42], making possible the production of a fully lignin-based composite. The industry is more focused on the use of lignin and cellulosic fibers as a source of reinforcement for the automotive and construction sectors through the fabrication of green composites, hybrid biocomposites and even biotextiles [43].

Another relevant compound used in this study is a biorefinery-derived side-product: humins, which are generated in the production of furandicarboxylic acid [44,45,46,47]. Humins are mainly composed of aromatic rings, containing different functionalities which vary depending on the biorefinery processing: aldehydes, hydroxyls, carboxylic and ketones. These reactive groups make the humins a promising building block and an important compound for the development of thermoset resins [37,44,45,47,48,49].

The objective of the present study is to prove the concept for the validation of the designed sustainable composites based on ELO, humins and natural fibers to be used in a key sector, the construction sector, as described in the frame of LIFE15 ENV/BE/000204 project “RECYSITE”. In this project, different alternatives of bio-based composite materials were studied for their application in the construction sector as non-structural parts. With the research shown in this manuscript, the authors also intend to approximate the research conducted during the last years and within the project to the market through the use of market-established and legally acceptable characterizations. Enabling market engagement to bio-based composites.

In order to provide all the information currently required by the market in the methodology, it is key to tackle the end-of-life of the proposed materials/products. Nowadays, as plastics are becoming a challenge for waste management, especially those involving composite materials, these materials cannot be melted due to the crosslinking of covalent bonds that link network chains, hindering recycling. Therefore, most of the waste of thermoset composites, from production or end-of-life, is not currently properly recycled (incinerated 41.6%; landfill 27.3%). Nevertheless, new strategies are being researched to propose new ways that enable to recycle the composite materials. 

## 2. Materials and Methods

### 2.1. Materials

The epoxidized linseed oil (ELO; average molecular weight = 980 Da; average functionality = 5.5 epoxides per triglyceride, viscosity ~1200 Pa.s) was purchased from Valtris Specialty Chemicals (Independence, OH, USA). CAPCURE^®^ 3-800 is a mercaptan-terminated product used as a liquid curing agent with unique rapid-cure characteristics for epoxy resins at ambient temperatures supplied by Gabriel Performance Products (Akron, OH, USA). The humins were kindly provided by Avantium (Amsterdam, The Netherlands). 

The non-woven hemp and linseed fibers were kindly supplied by CENTEXBEL (Gent, Belgium). All the different consumables used during the infusion process were purchased from MEL composites (Barcelona, Spain). 

### 2.2. Resin Composition and Preparation

The resin was created by combining the two comonomers, humins and ELO, with the hardener, Capcure 3-800, in different ratios, as given in Table 1.

A comprehensive study of systems reactivity was previously carried out by the dynamic DSC to reach these formulations, evaluating the enthalpies of polymerization reactions and also the interval of the reactions’ temperatures [48,49]. An important objective of this work was the valorization of humins; therefore, it was used as the primary monomer with the highest possible ratio (≥50%). For this reason, the selected formulation to be studied to prepare composites with hemp and linseed fibers was the one containing 50% humins, 30% Capcure and 20% ELO, as this formulation has a good reactivity and a bio-based content of ~70%. This formulation also has the advantage of good viscosity, allowing the fabrication of composites by liquid resin infusion. As shown in Figure 1, the crosslinking temperature of the resin is ~119 °C with a viscosity of the system of around 1 Pa·s (1000 cPs), with the lowest value of viscosity between 80 °C and 90 °C, where the viscosity decreases to ~900 cps.

To prepare the thermoset material, we heated the humins up to 75–80 °C, allowing them to flow and be mixed for 5 min with the corresponding quantity of ELO and under vigorous stirring with the Capcure at 75 °C. The final mixture was kept under continuous stirring for 10 min at 75 °C. Due to the slow polymerization rate, it was possible to proceed with the composites fabrication by liquid resin infusion (LRI) process and to maintain the composite for 3 h at 130 °C achieving complete curing [48]. 

### 2.3. Fabrication of Composites by Liquid Resin Infusion (LRI)

There are several technologies allowing the production of thermoset composite materials; some examples are pultrusion, resin transfer molding or liquid resin infusion (LRI), among many others [44]. LRI was chosen because of its versatility from a technological point of view and for its economic feasibility, allowing it to obtain a product that will be highly competitive in the market. The LRI processing technology consists of the generation of a vacuum in a covered mold where the reinforcement material has been previously placed and the use of the generated pressure gradient to pump the matrix into the mold until all the reinforcement is impregnated. At this point, the entrance is closed, and the vacuum is maintained through the curing process. Table 2 gives the compositions of 4 prepared composites with 1 or 3 fiber layers prepared. Once the infusion has been made, the resin is trapped and cured within the mold [36]. 

The process allowed the preparation of composites of ~0.16 m^2^, based on hemp and linseed fibers with different densities: 600 g/m^2^ (3 layers of reinforcement) and 1300 g/m^2^ (1 layer of reinforcement). Therefore, the final composites vary in terms of fibers and also in the amount of resin that has been incorporated, as summarized in Table 2.

### 2.4. Experimental Part 

#### 2.4.1. Thermal Conductivity

The thermal conductivity of the prepared materials was measured by the stationary methodology of the hot plate using a DTC-25 TA instrument (New Castle, UK). After reaching the thermal equilibrium, the temperature difference across the specimen is measured along with the output from the heat flux transducer. The thermal conductivity of the samples was determined by applying the equation:ΔQ/(Δt·A) = −k·(ΔT/Δx)(1)
where ΔQ is heat gradient (W); Δt is time gradient (s); A is the sample area (m^2^); k is the material constant thermal conductivity (W/(m·K)); ΔT is the gradient of temperature (K); Δx is the thickness of material (m).

#### 2.4.2. Tensile Testing

The Young’s modulus and the tensile strength of the linseed and hemp non-woven composites were measured with an Ibertest STIB-200/W–200 KN (Madrid, Spain) uniaxial test machine following ISO 527-5. As stated in the ISO, it used a 1 mm/s speed, and then the speed was increased to 5 mm/s. The tests were performed with 8 specimens of 17.5 cm length × 2.5 cm width from each composite, and the values were averaged. The manufacture of the specimens was performed through the LRI of a single plaque. It was afterward cut with a saw to obtain the specimens with a geometry of 2.5 cm width and 17.5 cm length; the thickness of the specimens is defined in Table 2. Taps made of glass fiber were incorporated at the top and bottom of the specimens with a standard epoxy resin in order to ensure a proper load transfer from the tensile machine to the specimen by using the same epoxy resin of the taps as adhesive between sample material and tap. 

#### 2.4.3. Flexural Testing

The flexural modulus of the composites was determined by using an Ibertest STIB-200/W–200 KN (Madrid, Spain) uniaxial test machine in three-points bending test method following the ISO 178 at a 1 mm/s and 2 mm/s speed. The tests were performed on 8 specimens for each composite of 15 cm length, 5 cm width and a distance between supports of 10 cm, and the values were averaged (Figure 2). The thickness of the specimens is defined in Table 2**.** The composite’s flexural modulus was calculated by using the equation: E_f_ = (Δσf)/(Δεf)(2)

E_f_ is the flexural modulus (MPa); Δσ_f_ is the variation in the flexural effort (Mpa); Δε_f_ is the variation in the deformation (%). 

#### 2.4.4. Cataplasma Test

The cataplasma test was used to study the aging behavior of the materials by measuring the physical changes caused in the samples (weight, shape and thickness) after their exposition to elevated temperatures and high humidity. The measurements were performed with Mirta-Kontrol equipment (Zagreb, Croatia) using ISO cataplasma test DIN EN ISO 9142. The samples were kept under 100% humidity and 70 °C for a month, and the measurements were taken after 1, 7, 14 and 28 days. Before performing the flexural test (described in Section 2.4.3), The samples were recuperated after 28 days and dried for 12 h at 60 °C. This drying process was applied in order to understand the aging effect, simulated by the high humidity conditions to which the samples have been exposed. In this way, two sets of flexural measurements were differentiated: one after the exposition to the conditions defined in the cataplasma test, after cataplasma (AC), and one previous to the exposition, before cataplasma (BC). Hence, 6 different specimens were tested with dimensions 15 cm of length × 5 cm of width (1 before cataplasma, 4 during cataplasma and 1 after cataplasma). 

#### 2.4.5. Scanning Electron Microscopy (SEM)

SEM was used to investigate the adhesion between the natural fibers and the humins-based thermoset matrices and the morphology of the fracture surface of the biocomposites. Fresh fractures of composites samples were mounted on an SEM stub and coated with platinum prior to observations. The instrument used is a Tescan Vega 3 XMU SEM (Prague, Czechia) at an accelerating voltage of 5 kV. 

#### 2.4.6. Fire Resistance Test

To test the behavior of the samples under fire conditions, we used Mirta-Kontrol equipment (Zagreb, Croatia). The reaction to fire tests method followed EN ISO 11925-2, in which the samples are subjected to a direct single-flame source. The analyses took place inside a test chamber where the specimen was mounted vertically and subjected to a gas flame. During the test, time of ignition, burning droplets and whether the flames reach certain mark (defined by the ISO) within a prescribed time period are registered. In this way, it is possible to evaluate fire retardancy and generation of burning drops. Three specimens (25 cm length and 9 cm width) were tested for each sample.

## 3. Results and Discussion

The first step towards the validation of composite material is the quality of adhesion between the reinforcement and the polymeric matrix. SEM analyses were used to observe and evaluate the interaction between both components. The micrographs presented in Figure 3 confirm the good adhesion between the natural fiber and the bio-resin. According to the SEM micrographs, the fibers’ surface is completely covered by the resin, proving the excellent compatibility between the two components of the biocomposite, the matrix and the biofiller. This is due to the excellent compatibility of vegetal fibers and the humins (hydrophilic macromonomer) as a consequence of their chemical affinity [24]. In Figure 3c–e, it is possible to observe that the fibers that come out of the matrix are impregnated with the resin, showing a good adhesion and compatibility between the fiber and the matrix. Moreover, on the fractured composite (Figure 3a), a good homogeneity of the fibers distribution can be noticed; moreover, a fibrillar breaking of the composite can be observed. That is due to the fact that the fibers are in a non-woven structure; therefore, once the matrix reaches its limit force, the fibers start to break. Meanwhile, it is observed that after breaking, the fibers are completely covered in the matrix, meaning that the adhesion between the fiber and the resin is stronger than the force that the matrix is able to withstand. This observation strengthens the previous observation on the adhesion between the fiber and the matrix. This is a key element in the development of a bio-based resin and in its success in a market application, as no treatment was given to the fibers to enhance the interaction.

### 3.1. Composites Properties

The different biocomposite specimens were fully characterized following the conditions previously described. 

#### 3.1.1. Thermal Conductivity 

For the thermal behaviour of the composite materials, firstly, the thermal conductivity of the natural fibers was measured. These measurements showed that the fibers’ thermal properties were in the same range as many other materials typically seen as good insulators, i.e., sheep wool and glass wool, with heat transmittance values of around 0.04 W/mK. It was found that natural fibers of hemp and linseed have similar values of heat transmittance capacity and that the density of these non-woven structures does not affect the final capacity of the sample to transmit or isolate the heat. The four composite samples have higher values of thermal transmittance compared to those of the neat fibers. These are multiplied by 2.5, reaching values of around 0.1 W/mK. This increment in the heat transmittance properties places the final composite materials in the range of other natural structures such as pine wood [50]. This result clearly shows the effect of the resin on the properties of the composite and also shows that the thermal conductivity of the resin is higher than one of the reinforcement materials. 

With the data shown in Figure 4, it is possible to appreciate that the final properties of these composites are in line with other natural materials typically used in building structures. Although the densities and the source of the reinforcement are different, the values obtained for all four different composites are in the same range. Therefore, we can evaluate these composites made from natural reinforcement and bio-based matrices as a new possibility for the development of products with isolation properties. However, in order to generate the best quality composite in terms of isolation properties, it is important that the amount of matrix is kept to a minimum as the main insulator material of the composite is the reinforcement: the natural fibers.

#### 3.1.2. Mechanical Properties

The developed biocomposites have been tested mechanically by two different methods: tensile and flexural tests.

##### Tensile Tests

The tensile strength and the elastic modulus have been studied by comparing the prepared composites from both types of fibers. In these comparisons, it is observed that the composite materials in which linseed fibers were used as reinforcement have a better mechanical performance than the hemp-based composites. This is aligned with the data reported by Padney et al. [51]. The research showed that linseed is one of the most competitive natural fibers and that it can even be compared to glass fiber. While comparing the tensile strength of linseed and hemp fibers, it is possible to observe that the linseed shows twice the strength value of the hemp at both densities and almost three times the elastic modulus of the hemp. Additionally, a non-significant difference is found depending on the fiber density, this difference is dependent on the type of fiber as it is not possible to observe it clearly in hemp (A, B), but it is significant for linseed (C, D). In the case of the tensile strength, the difference between the density of the two types of non-woven structures, triggered a drop of the tensile modulus value of a few MPa (more drastic in the case of the linseed), and for the elastic modulus, a drop of 100 Mpa in the linseed case (Figure 5). 

The comparison of the data results obtained with the reported tensile properties of woven reinforced composite materials (those gathered by Padney et al. [51] and Del Borello et al. [12]) show that the non-woven structure cannot compete with the woven structures due to the lack of directionality of the fabrics. As it is shown by Bochnia et al. [52], the direction of the reinforcement has a direct impact on the mechanical properties, and this effect is the same for all kinds of composite materials. The spread of results shown in the research, depending on the orientation, does not have any influence on these samples, as the fibers are randomly dispersed in a non-woven structure. These points imply that the non-woven-based composites can be used in any position as there is no preferred direction, and the mechanical measurements shown in Figure 5 are met in all of them. 

##### Bending Tests

The bending test results are shown in Figure 6. They display a similar behavior as those reported for the tensile properties. The linseed fibers have a higher mechanical performance than the hemp fibers, and high-density composites show higher bending modulus values. These results agree with the previous data illustrated in Figure 5, in which the linseed has already shown higher mechanical performance than hemp. Nevertheless, it is remarkable that in the case of high-density hemp composites (A), the bending modulus is in the same range as the low-density linseed composite (D). The smaller difference between both types of reinforcements in the bending tests in comparison with the values shown for the tensile properties is related to the matrix properties. As the mats are oriented perpendicularly to the force applied in the bending test, the bio-based matrix takes a bigger role in the bending properties and equalizes the values obtained for both types of composites. However, it must be remarked that even in this case, linseed fibers composites show better mechanical performance than hemp reinforced.

##### Cataplasma Tests

Physical changes in the composite specimens were measured during the exposure time. As specimens absorbed water, physical changes in weight, width and thickness were monitored (Figure 7, Figure 8 and Figure 9).

As shown in Figure 7, Figure 8 and Figure 9, the dimensions of the hemp-based composites increased up to 35% in weight and up to 15% in thickness during the first 24 h of exposure, while their width increased ~1% compared to the initial value. 

In contrast, the weight increases during the first week for the linseed were around 20%, whereas, for the hemp, the increment goes up to 40% of its initial value. At the same time, the width and thickness remain equal. Observing all these data and behaviors, we can say that in the case of both types of natural fibers, the absorption of all the water is completed after one day of exposure.

In Figure 10, values for the bending modulus for both composites before and after the exposure are compared. After the exposure, the low-density fiber-based composites have increased the values of the bending modulus by ~100 MPa, while in the case of high-density fiber-based composites, the bending modulus decreases by ~200 MPa, ending both structures with similar bending moduli. The only thing about the bending modulus that does not change is the fact that the linseed fibers still have values of around 550 MPa after the exposure, while for the hemp fibers, the values go down to 150 MPa. This implies that the linseed fibers are going to absorb similar amounts of water to the hemp, but due to their higher mechanical properties, they will still end up having a higher mechanical performance. The cataplasma test demonstrates that after the composite is saturated with water, the mechanical performance of natural linseed fibers results in a bending modulus four times higher than the hemp, which means 400 MPa higher. To improve the results and reduce the impact of water absorption on the final properties of the material, one could use different alkali-based surface treatments [26].

#### 3.1.3. Fire Resistance Test

All composite materials produced were tested for both their flame retardancy and the generation of drops. These tests show that none of the materials generate burning drops while they are on fire. 

Regarding flame retardancy properties, composite materials have shown a lower capacity to spread fire in comparison with natural fibers. As it is shown in Table 3, the presence of the resin has a flame retardancy effect on the final specimen. Meanwhile, linseed fibers were not able to withstand any of the tests carried out. Hemp fibers withstood the small burner for 15 s. When the composite as a whole is characterized, linseed fiber samples are able to withstand the small burner for 15 s, passing from class F to class E. In the case of the hemp, it jumps to the next step where further characterization and techniques are needed in order to differentiate between class B, C or D. The pattern of experiments followed in classifying a material is represented in Figure 11. With the defined experiments, it is possible to classify all materials into the different categories used for market materials.

Seeing all these data and how the different elements affect the fire propagation and, therefore, the final classification of the materials in the fire propagation classes, it is easy to come to two conclusions after observing the data gathered in Table 3. On the one hand, the presence of the resin and the composite as a whole has higher fire resistance, making the progress of the flame slower than in those cases where no bio-resin is incorporated in the sample. On the other hand, the most remarkable point is that even though all-natural fibers are usually seen in the same way, expecting a similar fire behavior, hemp has demonstrated to have lower fire transference than linseed, favoring its use for this purpose. 

Regarding the propagation to other components through the generation of burning drops, it can be observed that none of the studied composites generate burning drops. This has a direct implication for the propagation of the fire toward other components or materials.

## 4. Feasibility of the Scaling up Process 

To demonstrate the feasibility of the fabrication of a 1 m^2^ composite, the LRI was upscaled to a mat of 1 m^2^ with both types of fibers, demonstrating the capacity to manufacture bigger composite parts. 

On the other hand, for comparison with commercial non-structural parts, a sandwich structure was manufactured with a PET core, as in Figure 12b. To manufacture the sandwich structure, firstly, a test piece of 0.16 m^2^ was fabricated to evaluate the best approach, which finally was a perimetral infusion process. For the large scale, a combination of strategies was used to reduce the impregnation time; the perimetral infusion with two entrances was supported by another four entrances within the geometry to speed up the impregnation process. The main difficulty of the sandwich structure was to ensure the temperature on both sides of the sandwich (75–80 °C) to enable the resin to flow through the mats and PET’s holes, joining all the components of the sandwich structure and avoiding possible delamination issues in the composite during its lifetime. Afterward, it was possible to scale up the fabrication to a piece of 1 m^2^ (Figure 12a). In the end, it was possible to manufacture several pre-industrial scale parts. These demonstrators have proven that the natural non-woven fibers can be used together with a bio-epoxy matrix to generate a sandwich structure similar to those commercially available for non-structural parts.

For the end of life of the composite, as mentioned in the introduction, incineration is not an option, as nothing would be recovered. Nevertheless, mechanical grinding is a plausible solution. In order to validate this approximation, some samples were ground and introduced in other composite materials as reinforcement. Nonetheless, for composites with a sandwich structure, the key point is the separation of the components. For this demonstrator, the thermoset and PET core were separated in a water bath, allowing each of the materials to reach its specific recycling flow. 

## 5. Conclusions

The transition towards biomaterials in the composite industry is a real need, together with the use of natural fibers as reinforcement. Within this research, the focus has been placed on non-woven natural linseed and hemp fibers, with different reinforcing mat densities, and in the application of a market focus research methodology. Composites were produced by using a bio-resin and combining a biorefinery side-product, humins, with an epoxidized linseed oil-based monomer. The SEM micrographs provided a solid base for the application of both structures (matrix and natural fibers) together in the composite, as no third element was needed to reinforce their interaction. 

Regarding the analysis of the obtained composites, the thermal conductivity properties compete with other typical materials used in the construction sector for both types without significant differences. The mechanical performance of linseed-based composites showed superior values compared to that of hemp-based composites. Nevertheless, the hemp-based composites exceeded the performance of linseed in terms of fire resistance, whereas hemp has shown better performance in reducing the times for fire propagation.

Hence, each of the non-woven mats (linseed and hemp) has its own advantages, making hemp fibers the best option for non-structural parts that need an insulation material and have a higher risk of fire. For linseed fibers, the applications should be directed to the higher mechanical performance of non-structural parts. It is difficult to select one over the other without knowing all requirements of the application, as each shows advantages over the other in different fields. Nevertheless, the methodology and characterization techniques used demonstrate the potential of a market-oriented methodology to bring developments in the field of composites closer to their applications. This has not only been demonstrated with the production of some large parts but also through the use of standardized systems.

## Figures and Tables

**Figure 1 materials-16-01283-f001:**
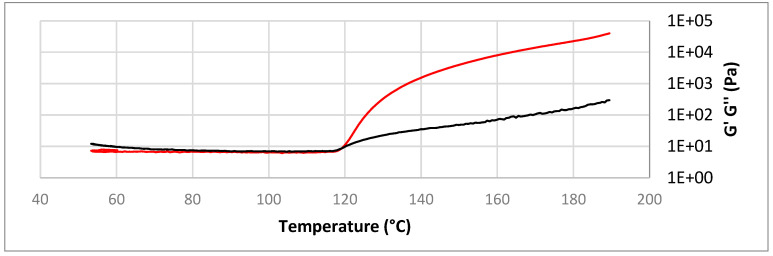
Evolution of storage (black) and loss moduli (red) with the temperature for the thermosetting resin 50%Hu/30%Cap/20%ELO.

**Figure 2 materials-16-01283-f002:**
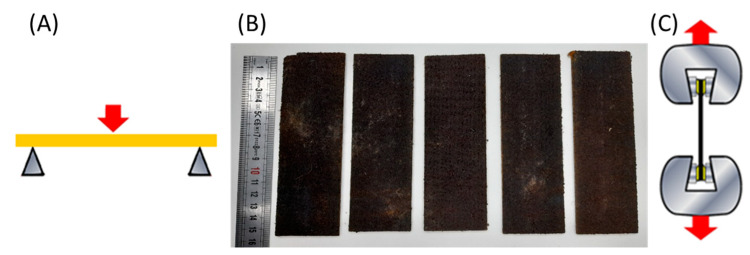
(**A**) Scheme of bending tests; (**B**) Non-woven composite specimens for bending test (**C**) Scheme of the tensile tests.

**Figure 3 materials-16-01283-f003:**
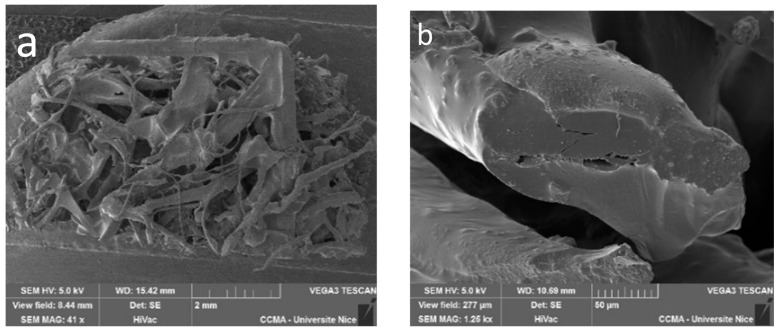
50%Hu/30%Cap/20%ELO and hemp 1300 g/m^2^ composites: (**a**) general look of the composite at a broken point; (**b**) single hemp fiber covered in resin; (**c**–**e**) single hemp fiber covered by the resin in three successive magnifications.

**Figure 4 materials-16-01283-f004:**
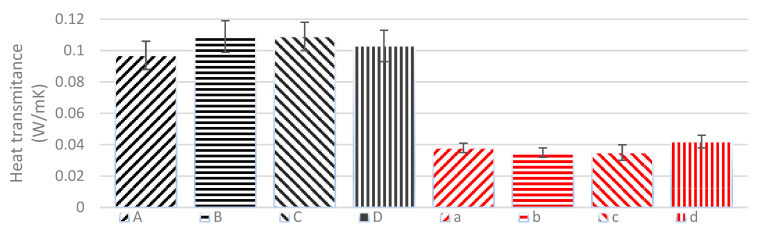
Heat transmittance of each of the different composites compared with the raw fibers (A: hemp composite 1300 g/m^2^; B: hemp composite 600 g/m^2^; C: linseed composite 1300 g/m^2^; D: linseed composite 600 g/m^2^; a: hemp 1300 g/m^2^; b: hemp 600 g/m^2^; c: linseed 1300 g/m^2^; d: linseed 600 g/m^2^).

**Figure 5 materials-16-01283-f005:**
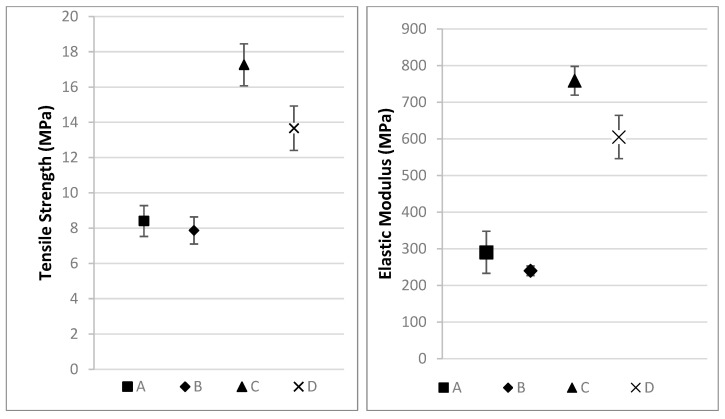
Tensile test results (A: Hemp composite 1300 g/m^2^; B: Hemp composite 600 g/m^2^; C: Linseed composite 1300 g/m^2^; D: Linseed composite 600 g/m^2^).

**Figure 6 materials-16-01283-f006:**
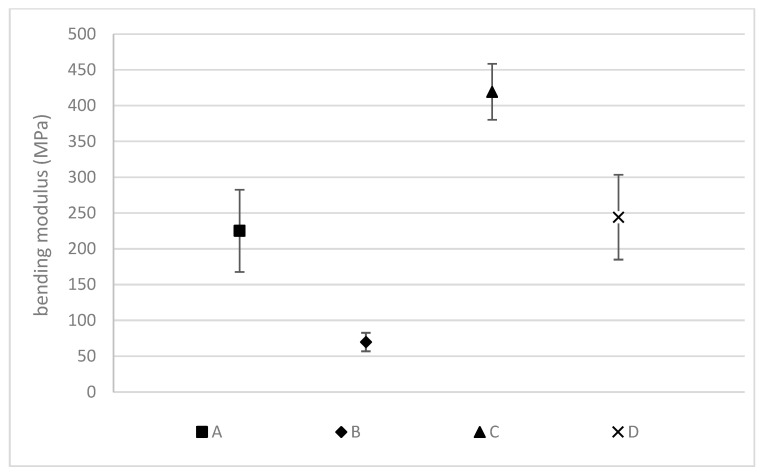
Bending tests (A: Hemp composite 1300 g/m^2^; B: Hemp composite 600 g/m^2^; C: Linseed composite 1300 g/m^2^; D: Linseed composite 600 g/m^2^).

**Figure 7 materials-16-01283-f007:**
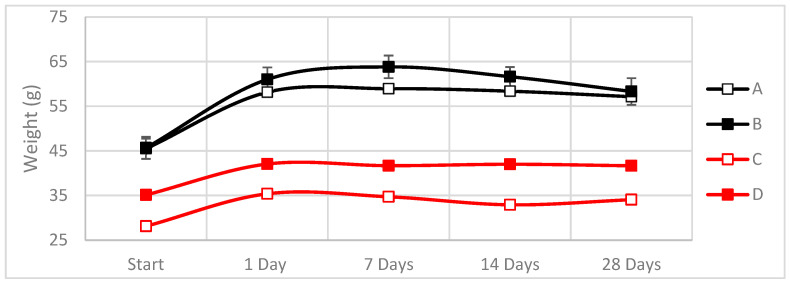
Variations in the weight during the cataplasma test due to the absorption of water (A: Hemp composite 1300 g/m^2^; B: Hemp composite 600 g/m^2^; C: Linseed composite 1300 g/m^2^; D: Linseed composite 600 g/m^2^). Raw data can be found in Table A1.

**Figure 8 materials-16-01283-f008:**
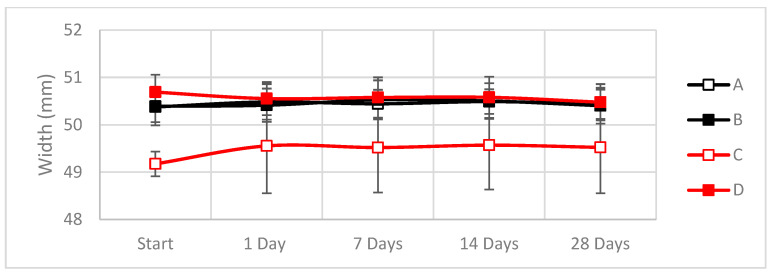
Variations in the width during the cataplasma test due to the absorption of water (A: Hemp composite 1300 g/m^2^; B: Hemp composite 600 g/m^2^; C: Linseed composite 1300 g/m^2^; D: Linseed composite 600 g/m^2^). Raw data can be found in Table A2.

**Figure 9 materials-16-01283-f009:**
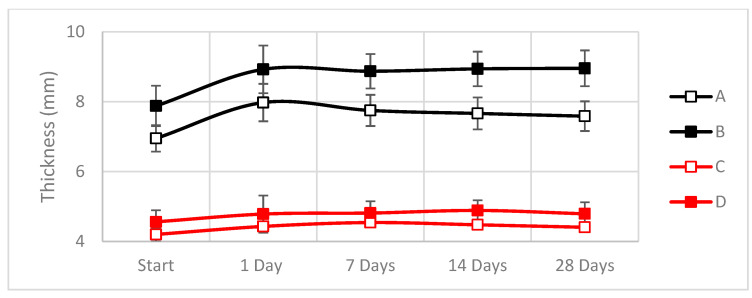
Variations in the thickness during the cataplasma test due to the absorption of water (A: Hemp composite 1300 g/m^2^; B: Hemp composite 600 g/m^2^; C: Linseed composite 1300 g/m^2^; D: Linseed composite 600 g/m^2^). Raw data can be found in Table A3.

**Figure 10 materials-16-01283-f010:**
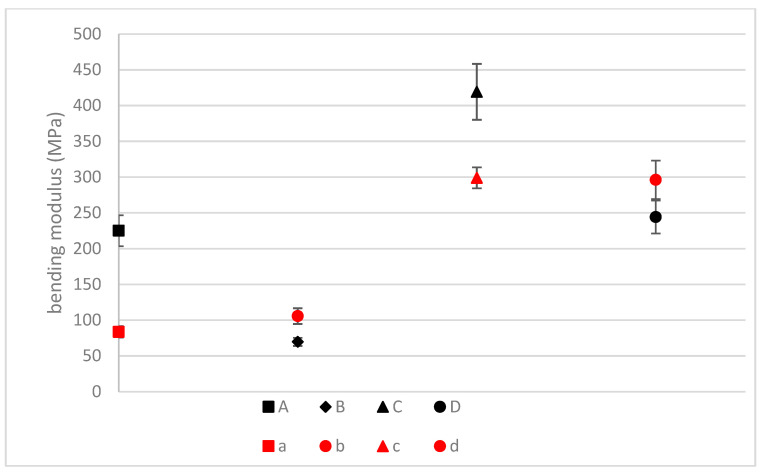
Variation of the mechanical properties before (BC) and after the cataplasma (AC) test (A: Hemp composite 1300 g/m^2^ BC; B: Hemp composite 600 g/m^2^ BC; C: Linseed composite 1300 g/m^2^ BC; D: Linseed composite 600 g/m^2^ BC; a: Hemp 1300 g/m^2^ AC; b: Hemp 600 g/m^2^ AC; c: Linseed 1300 g/m^2^ AC; d: Linseed 600 g/m^2^ AC).

**Figure 11 materials-16-01283-f011:**
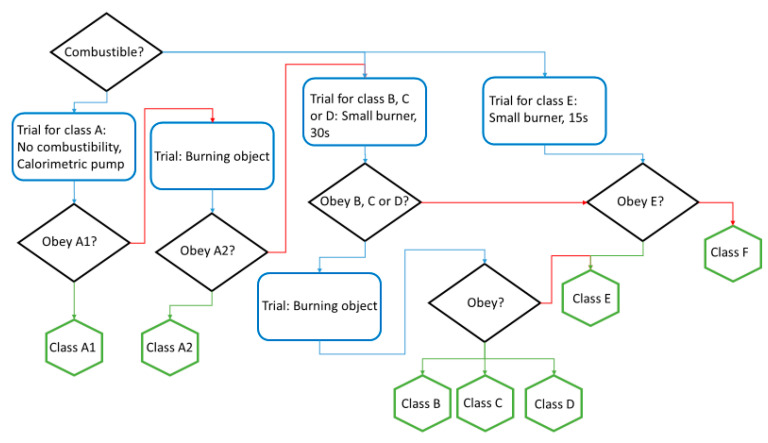
Scheme for the categorization of materials depending on their fire resistance.

**Figure 12 materials-16-01283-f012:**
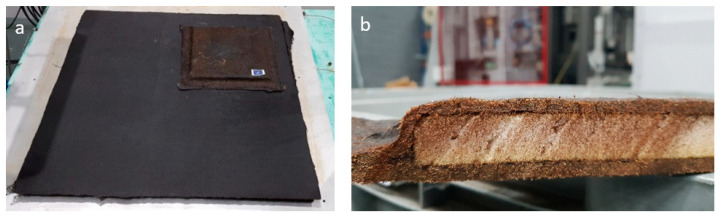
Size comparison between the demonstrators for industrial purposes (1 m^2^) and for material characterization (0.16 m^2^) (**a**); sandwich structure linseed-PET-linseed (**b**).

**Table 1 materials-16-01283-t001:** Compositions of designed formulations.

Humins (wt%)	Capcure 3-800 (wt%)	ELO (wt%)
30	52.5	17.5
40	45	15
50	37.5	12.5
50	30	20
60	30	10

**Table 2 materials-16-01283-t002:** Composites compositions based on different layers of hemp and linseed fibers.

Composite Based on 50%Hu/ 30%Cap/20%ELO	Fiber	Number of Layers	Fiber Density (g/m^2^)	Fiber Weight (%)	Resin Weight (%)	Thickness (cm)
A	Hemp	1	1300	29.9 ± 1.4	70.1 ± 1.4	5.8 ± 1.0
B	Hemp	3	600	31.1 ± 2.0	68.9 ± 2.0	7.1 ± 1.1
C	Linseed	1	1300	36.8 ± 1.3	63.2 ± 1.3	3.6 ± 0.5
D	Linseed	3	600	28.3 ± 2.3	71.7 ± 2.3	5.1 ± 0.5

Deviations shown in the table are between specimens. Each of the specimens’ thicknesses has a maximum deviation of 3%.

**Table 3 materials-16-01283-t003:** Fire resistance classification of bio-based non-woven epoxy composites.

Material	Classification
Hemp	E
1 Layer of Hemp 1300 g/m^2^	B, C or D
3 Layers of Hemp 600 g/m^2^	B, C or D
Linseed	F
1 Layer of Linseed 1300 g/m^2^	E
3 Layers of Linseed 600 g/m^2^	E

## Data Availability

The data presented in this study are available on request from the corresponding author. The data are not publicly available due to containing information that could compromise the privacy of research participants.

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
