# Peer review of "Sustainable Composites from Nature to Construction: Hemp and Linseed Reinforced Biocomposites Based on Bio-Based Epoxy Resins"

_materials, 2023, doi:10.3390/ma16031283_

Round 1
Reviewer 1 Report
The following comments are to be addressed while preparing the revised manuscript.
1. The paper requires a major English correction Most of the sentences are neither incomplete nor contains grammatical mistake. Hence, the reviewer would recommend a major English review by a native speaker or use the services offered by the journal.
2. Going through the introduction section, the novelty of the work is unclear. Hemp fibers are already used for the preparation of cementitious composites (Ex: 10.1016/j.matpr.2022.05.319). Please clarify the novelty of the present work. Moreover, the reviewer feels that the literature section is incomplete. A lot of recent papers similar to the above one is to be added so that the knowledge gap and the need for present work can be justified.
3. In addition to the above comment, the discussion in the paper is poorly detailed. The paper sounds more like a review article rather than a research article. Please improve the discussion of results in the manuscript.
4. Need for Figure 11 is unclear. Please clarify.
5. Overall, the conclusion section is weak and requires a major improvement. At present the conclusion section sounds like a mere summary.
Author Response
The authors thank the valuable comments of the reviewer of Materials and their time for revising our work. We have addressed all the comments mentioned and we answered all concerns related with our manuscript. We include a detailed answer to each of his comments and a list of the included changes.
Responses to reviewer #1
- The paper requires a major English correction Most of the sentences are neither incomplete nor contains grammatical mistake. Hence, the reviewer would recommend a major English review by a native speaker or use the services offered by the journal.
Dear reviewer we appreciate your comment regarding the English. In this regard, we have contacted a native colleague and she has done a thorough review of the article.
- Going through the introduction section, the novelty of the work is unclear. Hemp fibers are already used for the preparation of cementitious composites (Ex: 10.1016/j.matpr.2022.05.319). Please clarify the novelty of the present work. Moreover, the reviewer feels that the literature section is incomplete. A lot of recent papers similar to the above one is to be added so that the knowledge gap and the need for present work can be justified..
Dear reviewer, thanks for the comment and the information provided. We have included more literature to show all the previous work done in the field. Besides, we have enlarged the introduction in order to make clearer the motivation and objective of the research and its framework. You can find most of these modifications from line 125, with the introduction of a new paragraph.
- In addition to the above comment, the discussion in the paper is poorly detailed. The paper sounds more like a review article rather than a research article. Please improve the discussion of results in the manuscript.
Thanks for the comment. In order to provide a more detailed and clearer explanation of the results. SEM micrographs, thermal conductivity, mechanical properties and fire properties have been explained more in detailed. These inclusions together with the clearer English provides a higher level of detail and a better explanation of the results.
- Need for Figure 11 is unclear. Please clarify.
Dear reviewer, thank you for the comment. We have introduced a sentence to clarify the necessity of figure 11 in the text. We feel that it is important to show all the steps that a material has to go through for its validation and classification, for that we are presenting figure 11 as a summary and scheme of these steps.
- Overall, the conclusion section is weak and requires a major improvement. At present the conclusion section sounds like a mere summary.
Thank you for the recommendation. In this regard the authors have gone through the section and modified, clarify and add some more information. Although modifications have been done through all the conclusions sector. The major addition of information can be found in line 490
Reviewer 2 Report
The paper titled Sustainable composites from nature to construction Hemp and Linseed reinforced biocomposites based on biobased epoxy Resins prepared natural fiber reinforced biocomposites by liquid resin infusion method, showed the potential of both hemp and linseed natural fibers and their composites to be used as insulation materials. This paper proved that biomaterials are a suitable solution for developing sustainable products fulfilling the requirements of the end user applications by related tests. The article has some innovation, but there are still some problems that need to be explained or modified.
1. Are the plant fibers used in the manuscript treated with surface treatment? Such as silane treatment.
2. In abstract, we can not get any useful information, it is suggested that rewrite the abstract.
3. In introduction, it did not introduce clearly the questions to be studied and the innovation points of this paper.
4. “There is article that investigates the natural fiber reinforced biocomposites, suggest citing: Composite Structures. 2023;303:116313.
(http://dx.doi.org/10.1016/j.compstruct.2022.116313).
Composites Communications 2023, 37: 101448.
(https://doi.org/10.1016/j.coco.2022.101448)”
5. Figures 3 and 12 should be numbered (every figure), these are easier to understand.
6. Please keep the style of the table consistent. For instance, the Table 2 is obviously different to other tables.
7. The discussion of the results should be deeper.
8. The writing should be improved, eg., the tense should be consistent, the sentences should be polished.
Author Response
The authors thank the valuable comments of the reviewer of Materials and their time for revising our work. We have addressed all the comments mentioned and we answered all concerns related with our manuscript. We include a detailed answer to each of his comments and a list of the included changes.
Responses to reviewer #2
The paper titled Sustainable composites from nature to construction Hemp and Linseed reinforced biocomposites based on biobased epoxy Resins prepared natural fiber reinforced biocomposites by liquid resin infusion method, showed the potential of both hemp and linseed natural fibers and their composites to be used as insulation materials. This paper proved that biomaterials are a suitable solution for developing sustainable products fulfilling the requirements of the end user applications by related tests. The article has some innovation, but there are still some problems that need to be explained or modified. Proposes to supplement the abstract with one or two sentences on the final conclusions.
- Are the plant fibers used in the manuscript treated with surface treatment? Such as silane treatment.
Dear reviewer, thanks for the comment. In this case the natural fibers used in the article do not have any kind of surface treatment. In the frame of the project as a whole they were treated with different bases and acids. Nevertheless, as none of the treatments proven was significantly interesting it was decided to go with the natural fibers without treatment.
- In abstract, we can not get any useful information, it is suggested that rewrite the abstract.
We appreciate your observation and we have modified the abstract in order to provide a more in depth view of the work presented in the article, as well as the results obtained in it.
- In introduction, it did not introduce clearly the questions to be studied and the innovation points of this paper.
We appreciate the valuable comment of the reviewer. We have included more information in the introduction in order to make clearer the object of the research and its framework. You can find most of the modifications on line 125, with the introduction of a new paragraph.
- “There is article that investigates the natural fiber reinforced biocomposites, suggest citing: Composite Structures. 2023;303:116313. (http://dx.doi.org/10.1016/j.compstruct.2022.116313). Composites Communications 2023, 37: 101448. (https://doi.org/10.1016/j.coco.2022.101448)”
The authors thank the comment and we have included the information in the introduction as well as in the discussion of the results.
- Figures 3 and 12 should be numbered (every figure), these are easier to understand.
The authors thank to the comment and have modified the images accordingly to the suggestion. We have identified each of the images in the figure and related them in the text, so that the images could be more easily identifiable during the reading.
- Please keep the style of the table consistent. For instance, the Table 2 is obviously different to other tables.
Dear reviewer, thank you for the observation. The style of all tables has been homogenized, using the same format and size in all the tables.
- The discussion of the results should be deeper.
The authors thank the reviewer for the comment. We have gone more in depth in the explanation of the results obtained for the SEM micrographs, thermal conductivity, mechanical properties and fire properties. As well as, in the feasibility of the upscaling. In all of them changes have been made to provide a better explanation of the results
- The writing should be improved, eg., the tense should be consistent, the sentences should be polished.
Thanks once again for the comment. In this regard, we have modified the writing of the article as a whole through contacting a native speaker colleague who has gone through the whole article.
Round 2
Reviewer 1 Report
Comments were addressed. Paper can be accepted for publication.
Author Response
Dear reviewer, thank you for your feedback and recommendation. In order to keep improving the manuscript and the English language we have shared with a colleague from the EEUU the manuscript, who has assisted us to improve the style.
Thanks once again for considering our work for publication.
Looking forward hearing from you
With kind regards